# Synergistic Interactions between Tocol and Phenolic Extracts from Different Tree Nut Species against Human Cancer Cell Lines

**DOI:** 10.3390/molecules27103154

**Published:** 2022-05-14

**Authors:** Jazmín C. Stevens-Barrón, Abraham Wall-Medrano, Emilio Álvarez-Parrilla, Imelda Olivas-Armendáriz, Humberto Astiazaran-García, Ramón E. Robles-Zepeda, Laura A. De la Rosa

**Affiliations:** 1Department of Chemical-Biological Sciences, Institute of Biomedical Sciences, Autonomous University of Ciudad Juarez, Ciudad Juárez 32310, Mexico; ealvarez@uacj.mx; 2Department of Veterinary Sciences, Institute of Biomedical Sciences, Autonomous University of Ciudad Juarez, Ciudad Juárez 32310, Mexico; 3Department of Health Sciences, Institute of Biomedical Sciences, Autonomous University of Ciudad Juarez, Ciudad Juárez 32310, Mexico; awall@uacj.mx; 4Department of Physics and Mathematics, Institute of Engineering and Technology, Autonomous University of Ciudad Juarez, Ciudad Juárez 32310, Mexico; iolivas@uacj.mx; 5Department of Nutrition, Food and Development Research Center A.C., Hermosillo 83304, Mexico; hastiazaran@ciad.mx; 6Department of Chemical-Biological Sciences, University of Sonora, Hermosillo 83000, Mexico; robles.zepeda@unison.mx

**Keywords:** food antioxidants, antiproliferative, cell viability, pharmacological interactions, cancer cell lines, bioactive compounds, polyphenols, tocopherols, tocotrienols

## Abstract

Tree nuts are rich in polar (phenolic compounds) and non-polar (tocols) antioxidants, with recognized effects in the prevention of diseases such as cancer. These biomolecules possess antiproliferative activity on cancer cells; however, the combined effect of both types of compounds has been scarcely studied, and this approach could give valuable information on the real anticancer potential of tree nuts. In the present study, the antiproliferative activity of pure tocols and phenolic compounds, tocol- and phenolic-rich extracts (TRE and PRE, respectively) from tree nuts and the extracts combinations, was evaluated in four cancer (HeLa, MCF7, PC3, A549) and one control (*ARPE*) cell lines. The most sensible cell lines were HeLa and MCF7. TRE and PRE from nuts were chemically characterized; γ and δ tocopherols, total tocols, total tocopherols and total phenolic compounds were negatively correlated with cell viability in MCF7 cells. In HeLa cells, only δ and total tocopherols were negatively correlated with cell viability. TRE and PRE had a low effect in reducing cell viability of the cancer cell lines, the most effective extracts were those of emory oak acorn (EOA), pecan nut (PEC) and walnut (WAL), and these were further studied for their pharmacological interactions, using the combination index and the isobologram methods. Combinations of both extracts showed a synergistic and strongly synergistic behavior in the three nuts (EOA, PEC and WAL), with combination indexes between 0.12 and 0.55. These results highlight the need to understand the interactions among components found in complex natural extracts or food products in order to fully understand their bioactivities.

## 1. Introduction

Cancer is one of the leading causes of death worldwide with 10 million victims in 2020, according to the World Health Organization (WHO) [1]. Resistance to chemotherapy is one of the main causes why cancer still has many victims, this occurs some of the cancer cells develop a survival strategy and the therapy is ineffective. Scientific studies reveal that a good diet contributes to the effectiveness of chemotherapies. Certain foods such as grains, fruits and vegetables have been considered as preventives or adjuvants in the treatment of cancer patients, due to their content of bioactive compounds with anticancer activities [2].

Tree nuts are an example of this type of food [3,4,5]. Moreover, tree nuts are remarkable because, since they are composed of an oily fraction and a polar fraction, they possess both lipophilic and hydrophilic bioactive compounds. The oil fraction contains molecules from the tocol family, tocopherols (Ts) and tocotrienols (T3s), whose difference is in the degree of saturation of its side chain. Ts contain a saturated side chain and T3s an unsaturated side chain with three double bonds. There are 4 forms of both T and T3: α, β, γ, and δ, depending on the number and position of methyl groups in the chromanol [6]. Ts, mainly αT, are the main forms of vitamin E; they are effective antioxidants, and the γ and δ forms also possess anticancer effects [7,8]. They can modulate and inhibit estrogen receptors [7] and cyclin D1 [8], thereby blocking the phases of cell growth and proliferation. T3s are even more effective than Ts in modulating these antiproliferative targets and reducing cancer cell survival [9]. δ and γ T3 are involved in the direct activation of the tumor suppressor protein p53, which in turn modulates the pro apoptotic regulator BAX/BAK and inhibits anti apoptotic Bcl-2 proteins [10,11,12].

In the polar fraction of nuts, the main phytochemicals are phenolic compounds whose chemical structure consists of one or more benzene rings substituted by at least one hydroxyl group. These molecules are mostly recognized for their antioxidant properties, and the most common phenolic compounds in nuts are gallic, syringic, ellagic and chlorogenic acids, catechin, epicatechin and related flavan-3-ols [13]. Their principal mechanism of action in cancer cells is the generation of intracellular reactive oxygen species (ROS) by autoxidation [14], cell cycle regulation [15] and activation of pro apoptotic pathways [16].

Although the antiproliferative activity of these molecules in cancer cells has been proven, their effectiveness in complex mixtures, as they are found in nuts and other foods, could be altered. A pharmacological interaction is the reciprocal action between two (or more) drugs or active principles that may reduce (antagonism) or increase (synergism) their effect when they are combined, so the combination should be considered as a different drug [17]. Pharmacological interactions may occur between different components of a natural extract or food product, but they are seldom studied. Combination studies that aim to evaluate synergistic or antagonistic effects between tocols and phenolic compounds should be carried according to the experimental and mathematical conditions of the Chou–Talalay method [18,19]. Therefore, the objective of this work is to determine the type of pharmacological interaction that may exist between tree nut tocols and phenolic compounds with respect to their antiproliferative effect in cancer cells, following the Chou–Talalay method. For this purpose, seven tree nut species were selected. Six of them are commercial and commonly consumed as healthy snacks: almond (ALM), pecan (PEC), pine nuts from two varieties (pink (PNP) and white (PNW)), pistachio (PIS) and walnut (WAL). The last one is a wild emory oak acorn (EOA) species characteristic of Northern Mexico, which can be used as an ingredient in baking products. Since each tree nut species possess a unique profile of tocols and polyphenols, their antiproliferative potential will be compared among them. This study will help to understand if the presence of both non-polar tocols and polar and phenolic compounds in nuts could be one reason for the cancer-preventing properties of nuts.

## 2. Results and Discussion

### 2.1. Antiproliferative Activity of Pure Tocopherols and Phenolic Compounds in Cancer Cell Lines

Tocopherols and phenolic compounds have shown antiproliferative and pro-apoptotic properties in many cancers cell lines [20,21]. In accordance with this premise, all compounds tested in the present work showed antiproliferative activity in all cell lines (Table 1). Gallic acid, a phenolic acid, was the most effective and showed the best effect in PC3 prostate cancer cell line (EC50: 98.6 µM), and these values are higher than those reported by Heidarian et al. [22] and Saffari-Chaleshtori et al. [23], where they establish an EC50 of 50 µM and 35 µM, respectively. It is known that gallic acid can promote apoptosis in prostate cancer and other cancer cell lines by inducing oxidative stress [14,24] and activating the intrinsic and extrinsic apoptosis pathways [16]. Of the three tocopherols studied, δT was more effective and its effect stronger in the MCF7 breast cancer cell line (EC50: 319.9 µM), followed by γT, most effective in PC3 cells (EC50: 436.9 µM). This was expected, since it is generally admitted that αT has lower antiproliferative, proapoptotic and, in general, anticancer activity than γ and δ isoforms [25].

To our best knowledge, there are no studies that indicate the EC50 values of γT and δT in MCF7, there is an estimate reported by Figueroa, Asaduzzaman, and Young [26] where the mean effective concentration for γT was between 50 to 75 µM. As well as the studies by Hong et al. [27], where γT and δT presented a > 50% inhibition of cell growth at doses between 60 to 100 µM. No studies were found that determined the EC50 of tocopherols in HeLa cells, while the study by Jiang et al. [28] in PC3 and A549 cells also observed a dose response effect of γT, with EC50 values of 50 and 40 µM, respectively.

The mechanisms of action of tocopherols in cell lines include upregulation (increased expression) of PPARγ [29,30], cell cycle arrest by decreasing protein levels of cyclins D1 and E [30,31] and apoptosis induction by activation of the p53 tumor suppressor protein and caspases 9 and 3, which in turn activate apoptotic proteins such as BAX and inhibit anti-apoptotic proteins such as Bcl-2 [12,30]. Nevertheless, none of the natural compounds was as effective as the chemotherapeutic agent doxorubicin, whose mechanism of action involves the generation of free radicals that damage DNA, cell membrane and proteins [32].

### 2.2. Antiproliferative Activity of Tree Nut TRE and PRE

The profile of tocols (T and T3) in the TRE and content of total phenolic compounds in PRE are shown in Table 2 and Table 3, respectively. γT was the most abundant isoform in all TRE and was highest in EOA; EOA, PNP and WAL extracts were also rich in δT, while αT showed the lowest recovery in all TRE and was only moderately high in ALM. High levels of T3 were present in both pine nut varieties (PNP and PNW), and moderate levels in EOA and PIS (Table 2). The presence of T3 in pink and white pine nuts and in EOA has been recently reported [33] and is relevant since tree nuts are not usually considered good sources of these vitamin E forms, and only PIS has been previously recognized for its T3 content [34,35]. The highest content of phenolic compounds was found in PEC and WAL extracts, while PIS showed the lowest content (Table 3). Next, the effect of TRE and PRE obtained from six species of tree nuts on the viability of the same cancer cell lines studied with pure compounds, plus a non-cancer retinal epithelial cell line (*ARPE*) was examined. Antiproliferative activity of TRE against all cell lines is shown in Figure 1. In general, HeLa cells were more sensible to all extracts, followed by MCF7 and PC3; A549 lung cancer cells and the non-cancer line *ARPE* were not affected by the TRE. In the HeLa cell line, EOA and WAL were statistically the most effective extracts, in MCF7 cells, EOA, WAL and PNP were the most effective and in PC3, PNP, WAL and EOA were the most effective (*p* < 0.05, statistical analysis in Appendix A). Since *ARPE* cells were not affected by TRE, all extracts showed selectivity indexes (*SI*) greater than one in HeLa, MCF7 and PC3 cells (Appendix A); however, all *SI* values were lower than 3; the highest value was for EOA TRE in HeLa cells (*SI* = 2.33), which indicates that these compounds do not show high selectivity toward cancer cells [36]. Moreover, cell viabilities at the highest extract concentrations were all above 50%, except for EOA extract in HeLa cells, indicating a low antiproliferative effect of all TRE. This was most probably due to the relatively low tocol concentrations in the extracts; the highest total tocol content was found in EOA extract, followed by PNP, and WAL (see Table 2), which were also the most active extracts. Correlation analysis of tocol content in extracts (Table 4) and cell viability showed a significant negative correlation for γT (MCF7 and PC3 cells), δT (HeLa and MCF7), total T (HeLa, MCF7 and PC3) and total tocols (T + T3 in MCF7 and PC3 cells). This indicates that γT and δT were the most active tocol forms, which is consistent with the results obtained from pure compounds (Table 1) and other published studies [25].

The effect of PRE on cell viability is shown in Figure 2. It is noticeable that the first doses of all extracts increased the number of live cells with respect to control (DMSO-treated cells) in all cell lines. For the four cancer cell lines, viability was maximal at 200 µg/mL of most PRE, and for the *ARPE* cell line, viability continued to increase as the extract dose increased, reaching values of 120.9 (PIS) to 157.2 (PEC) % from control. Phenolic compounds are well known for their antioxidant and cell-protective activities; nonetheless, they may behave as pro-oxidant and cytotoxic at high concentrations (above 50 µM according to several studies) and in the presence of metals; their prooxidant activity and ability to induce mitochondrial dysfunction and consequently apoptosis has been suggested as one of their possible anticancer mechanisms [21]. Lower doses of phenolic compounds usually increase the viability of cells subjected to oxidative stress but have no effect in non-stressed cells; however, one study has demonstrated stimulation of proliferation in an osteoblastic cell line by pure phenolic compounds used at low micromolar concentrations and by phenolic extracts from different olive oil varieties [37]. It is not clear if, and how, phenolic compounds could possess mitogenic activity, but the possibility should be further investigated.

The highest doses of PRE used in the present work were capable to reduce cell viability in MCF7 and HeLa cells (except PNW extract in HeLa). The PEC extract was the most effective in the MCF7 line and WAL in HeLa cells, but differences were not statistically significant with the other active extracts. Nevertheless, a significant negative correlation was observed in MCF7 cells between the extracts, total phenolic content, and cell viability (Table 4). Previous studies that have found antiproliferative activity of tree nut phenolic extracts have used extract concentrations in the order of mg/mL [38], so it is reasonable to think that PRE doses used in the present work were too low to exert a robust antiproliferative effect. Comparison of PRE and TRE antiproliferative effects indicated that TRE were more effective, although reduction in cell viability was still under 50% for all TRE treatments but one. Therefore, we decided to combine both extracts and analyze the antiproliferative activity of the mixtures to determine if any kind of pharmacological interaction could exist between both types of bioactive compounds. For this, we selected the most active extracts, EOA (most active TRE), PEC (most active PRE) and WAL (good activity of TRE and PRE), and the most sensible cell lines, MCF7 and HeLa.

### 2.3. Analysis of TRE and PCE Combinations on HeLa and MCF7 Cell Viability

Combinations of TRE and PRE were assayed by adding a sub-optimal concentration (EC20) of each extract to different concentrations of the other extract. The effect of all combinations in HeLa cells are shown in Figure 3, and their effect in MCF7 cells are shown in Figure 4. Calculated EC20 and EC50 values for all treatments and combinations are shown in Appendix A. Extracts from all nuts (EOA, PEC and WAL) showed the same tendency in both cell lines. TRE + PRE_EC20_ had a greater effect in reducing cell viability than the TRE alone, EC50 values of all TRE in combination with PRE were lower than 800 µg/mL, unlike when tested alone on cells, this means that combined extracts are more effective than extracts alone. For the combinations PRE + TRE_EC20_ the effect was more remarkable in HeLa and MCF7, the EC50 of the combinations always remained below 500 µg (Appendix A), and the combinations were more effective than PRE alone [EC50 above 2.0 mg (Appendix A); moreover, combination treatments did not induce an increase in viability at low PRE concentrations (200 µg), as observed when PRE were used alone (Figure 3 and Figure 4). These data suggested that PRE and TRE had a synergic effect on cell viability. To confirm the synergic interaction, the combination index (CI) was calculated for each experiment and data were also analyzed by the isobologram method. Figure 5 shows the isobolograms obtained from the combinations of EOA extracts in HeLa an MCF7 cell lines. In all cases, the combination appeared in the lower part of the isobologram, indicating a synergistic effect. The other extracts showed the same tendencies, which can be also observed in the CI values (Table 5).

The values of the CI were classified following the parameters of the Chou-Talalay method [19], where a CI between 0.1 and 0.29 indicates strong synergy and a CI of 0.3 to 0.69 indicates synergy. The combination PRE + TRE_EC20_ of all nuts presented values below 0.29 in both cells, which indicates strong synergy. In the combination TRE + PRE_EC20_ all CI values showed synergy between extracts (values between 0.37 and 0.55) in both cell lines.

The results consistently showed a synergy between both extracts; moreover, when a suboptimal concentration of TRE was added to PRE (PRE + TRE_EC20_) the interaction was strongly synergic for all nut extracts in HeLa cells and PEC and WAL extracts in MCF7 cells. A synergic interaction between tocols and phenolic compounds has been documented using pure compounds. In two studies, the antiproliferative activity of δT3 was potentiated by ferulic acid [39] and the lignan sesamin [40]. In both works, the authors concluded that the mechanism for the synergy was that the phenolic compound decreased the intracellular degradation of δT3 by inhibiting the activity of cytochrome P450 (CYP450), and thus the intracellular concentration of active δT3 was effectively increased [39,40]. CYP450 4F2 is involved in tocol side chain degradation, which is initiated by hydroxylation catalyzed by CYP450 families in the cell membrane [41]. Therefore, the synergistic interaction between our extracts could be mediated by the inhibition of CYP4F2 by the polyphenols in PRE and a decrease in the degradation of the tocols in TRE, which increases tocol bioavailability and its capacity to reduce cell viability.

Furthermore, a direct antiproliferative action of the PRE cannot be excluded. It is possible that tocols and phenols from the extracts modulate in synchrony different antiproliferative or cytotoxic pathways, such as the inhibition of mitogenic proteins (cyclins D1 and cMyc) and activation of caspases (3 and 9) by tocols [8], and a pro-oxidant activity and inhibition of mitogenic signaling (NF-kβ, MAPK, Cdk2-4) by phenols [42].

## 3. Materials and Methods

### 3.1. Samples and Reagents

Raw tree nuts of 6 species (5 commercial and 1 wild) were used in this study. Commercial species were almonds (ALM; *Prunus dulcis*), pecans (PEC; *Carya illinoinensis*), pine nuts (*Pinnus cembroides*) of pink (PNP) and white (PNW) varieties, pistachios (PIS; *Pistacia vera*), and walnuts (WAL; *Juglans regia*). They were all purchased from local retailers. Emory oak acorns (EOA; *Quercus emory*) were collected from wild trees located in Cuitaca, Sonora Mexico (31°00′17″ N, 110°29′33″ W) in the period September–October 2018.

Pure (≥93%) standards (gallic acid, α–, γ–, and δ–tocopherol); doxorrubicin, Folin–Ciocalteau phenol reagent, dimethyl sulfoxide (DMSO), Dulbecco’s modified Eagle’s medium, fetal bovine serum (FBS), 3- (4,5-dimethylthiazol-2-yl) 2,5 diphenyltetrazolium bromide (MTT), sodium chloride (NaCl), sodium hydroxide (NaOH), trypan blue, trypsin-EDTA, sodium phosphate salts were purchased from Sigma-Aldrich (St. Louis, MO, USA). Non-essential amino acids, L-Glutamine (200 mM), penicillin-streptomycin from Gibco (Gibco, Austria). High-performance liquid chromatography (HPLC) and analytical-grade solvents (hexane, methanol, acetone, and isopropyl and isobutyl alcohol) were obtained from JT- Baker (Avantar Performance Materials SA de CV, Mexico).

### 3.2. Preparation and Characterization of Tocol- and Phenolic-Rich Extracts (TRE, PRE)

All nuts were ground, oil was extracted by cold extraction method [43], and 10 g of grounded nut were mixed with 100 mL of hexane at 8000 rpm for 3 min. The mixture was filtered, the residue was re-extracted twice, the solvent portions were combined, and the hexane was removed by rotary evaporation (BUCHI Series-114, BUCHI Labortechnik AG, Flawil, Switzerland) at 40 °C. Oil was weighed and transferred to an amber bottle, sealed with nitrogen, and stored at −80 °C until analysis.

Tocols were then extracted from the oil fraction, and phenolic compounds from the defatted flour. The tocol-rich extract (TRE) was prepared according to Miraliakbari and Shahidi [43], with minor modifications. An amount of 20 g of oil was mixed with 200 mL hexane in a separatory funnel, then 100 mL of MeOH was added and the separating funnel was sealed and stirred for 15 min, and the methanolic fraction was recovered in a flask. The extraction process was repeated four times, all the methanolic fractions were mixed and the solvent was removed by rotary evaporation.

The quantification of individual tocols (α, γ and δT, α and γT3) in TRE was performed by normal-phase HPLC (Perkin Elmer model 200) according to the chromatographic conditions and methodology of Stevens et al. [33] and the results were expressed in µg tocols per gram of extract (µg tocols/g of TRE). For the phenolic rich extract (PRE), 1 g of defatted flour was mixed with 10 mL of 80% aqueous acetone in an ultrasound at room temperature for 10 min, and the mixture was centrifuged for 10 min at 3000 rpm at 4 °C. Next, the supernatant was recovered, and the sample was subjected to a further extraction process under the same conditions. The supernatants were combined, the solvent was removed by rotary evaporation, and water removed by lyophilization (LABCONCO Freezone 6, Labconco, Kansas City, MO, USA). Total phenolic compounds were quantified in the extracts according to De la Rosa et al. [44] and the results were expressed in µg equivalents of gallic acid/g of extract (µg GAE/g of PRE).

### 3.3. Cell Cultures

Cancer cell lines MCF7 (breast cancer), HeLa (human cervical carcinoma), A549 (human alveolar carcinoma), PC-3 (prostate adenocarcinoma), and noncancerous *ARPE* (endothelial retinal) were purchased from the American Type Culture Collection (ATCC, Rockville, MD, USA). Cells were cultured in Dulbecco Modified Eagle’s medium (DMEM) supplemented with 10% (*v/v*) fetal bovine serum, non-essential amino acids (7.5 mL), 10% L-Glutamine (200 mM), 100 U/mL penicillin and 0.1 mg/mL streptomycin (Gibco, Austria) under 5% CO_2_ and 95% air atmosphere at 37 °C. Cells were grown in 25 cm^2^ plastic flasks.

### 3.4. Antiproliferative Activity of Pure Compounds and Extracts

Cells were removed with trypsin-EDTA, seeded into 96-well plates (100 μL/well) at a density 1 × 10^4^ cells/mL and incubated for 24 h. Pure compounds (α, γ and δT, gallic acid, and doxorubicin), TRE and PCE extracts were initially dissolved in DMSO to prepare a stock solution and further diluted with DMEM to obtain the appropriate final concentrations in the microplate (1–1000 µM for pure compounds, 100–800 µg/mL for the extracts). DMSO was used as control with a final concentration of 800 µg/mL. The cell viability assay with the MTT reagent was performed according to Meneses-Sagrero et al. [45], 50 µL of DMEM with treatments or DMSO were added and incubated for 48 h. After the incubation period, cells were washed with 100 µL phosphate-buffered solution (PBS), then 100 µL of DMEM with 10 µL of MTT (5 mg/mL) was added to each well and later it was left incubating for 4 h more. The formazan crystals were dissolved with acidic isopropyl alcohol (where 337 µL of HCL are dissolved on 100 mL of isopropyl alcohol), and the absorbance of the samples was read at 570 nm in a UV-Vis microplate reader (BioRad Benchmark Plus) and a wavelength reference of 630 nm. Cell viability was expressed in terms of percentage, where the optical density of cells treated with only DMSO was considered as 100% viability.

EC50 (half effective concentration) and EC20 (20% effective concentration) values were determined by linear regression between % viability vs. log concentration [19]. A modified selectivity index (*SI*, Equation (1)) was determined for the extracts using the viability at the maximum concentration of extract (800 µg/mL):(1)$$SI=\frac{\%\;cell\;viability\;noncancerous\;cell\;line\;\left(ARPE\right)}{\%\;cell\;viability\;cancer\;cell\;line\;}$$
where *SI* > 1 means that cytotoxicity for cancer cells exceeds cytotoxicity in normal cells, and a value greater than 3 is considered as a high selectivity and safe for normal cells [36].

### 3.5. Experimental Design for Combinations and Analysis of the Pharmacological Interaction between TRE and PRE

TRE and PRE were combined and its antiproliferative effect determined as previously described. Two types of combinations were carried out: TRE + PRE_EC20_ and PRE + TRE_EC20._ In both, a sub-effective (EC20) concentration of each extract (extract 2 in Equation (2)) was mixed with different concentrations (100–800 µg/mL) of the other extract (extract 1 in Equation (2)), both from the same nut. Once the combinations were calculated and the extracts mixed and diluted with DMEM to the appropriate concentrations, the cell viability was determined with MTT as described in the previous section.

The combination index (*CI*) was calculated according to the Chou-Talalay method to determine the kind of interaction between TRE and PRE [19]:(2)$$CI=\frac{D1}{E1}+\frac{D2}{E2}$$
where *D*1 and *E*1 are the doses of extract 1 necessary to elicit the same effect and *D*2 and *E*2 are the doses of extract 2 necessary to elicit the same effect. *E*1 and *E*2 were calculated with each extract used alone in a different experiment, and *D*1 and *D*2 were calculated for each extract in the combination experiment. Considering the experiment design, *D*1 and *E*1 were EC50 values of extract 1 alone or in the combination, respectively; *D*2 was the suboptimal concentration (EC20) of extract 2 (since in the combination experiment, this extract concentration was able to elicit a 50% viability when combined with the appropriate concentration of extract 1) and *E*2 the EC50 of extract 2 alone. A *CI* of 0.90 or lower indicates synergy between extracts, 0.91 to 1.10 an additive effect, and 1.11 or greater means an antagonistic effect.

Interactions were also evaluated by the isobologram method, which is related to Loewe’s additivity [17]. In this graphical procedure, the doses of one drug (extract 1) are shown along an Y axis, and the doses of a second drug (extract 2, the one that potentiates the effect of extract 1), are shown along the X axis. Then, on each axis, the EC50 value of each extract alone is plotted (*E*1 and *E*2) and both points are joined by means of a line (isobole). Finally, a point is drawn in which *x* corresponds to the concentration of extract 2 in the combination that elicits a 50% viability (*D*2, which according to our experimental design will be the EC20 of extract 2) and *y* is the concentration of extract 1 in the combination that elicits a 50% viability (*D*1). The kind of interaction is deduced from the position that the point (*D*2, *D*1) occupies with respect to the isobole: if the point is below, the interaction is synergistic, close to the isobole is additive interaction (or no interaction), and above the isobole it indicates antagonistic interactions.

### 3.6. Statistical Analysis

All analyses were carried out three times in triplicate, and assays data were expressed as mean ± standard error of the mean (SEM). The data were subjected to a one- or two-way ANOVA, followed by a post hoc Tukey test to evaluate difference between samples (tree nut species) and between effects in different cell lines. A value of *p* < 0.05 was considered statistically significant.

## 4. Conclusions

This study indicates that although pure tocols and phenolic compounds can decrease the viability of cancer cells, phenol- and tocol-rich extracts (PRE and TRE) from tree nuts (known for their abundance in both types of antioxidants) had low antiproliferative activity, especially PRE. However, combination of both extracts (PRE + TRE) had greater effectiveness, showing strongly synergistic interactions. This finding highlights the importance of studying and understanding the interactions that occur among the multiple chemical components of complex natural extracts and food matrixes in order to fully understand their biological activities and health-promoting potential. More studies are still required to determine the mechanism of the synergy between tocols and polar phenolic compounds in the modulation of the pathways responsible for cell viability, proliferation, or apoptosis.

## Figures and Tables

**Figure 1 molecules-27-03154-f001:**
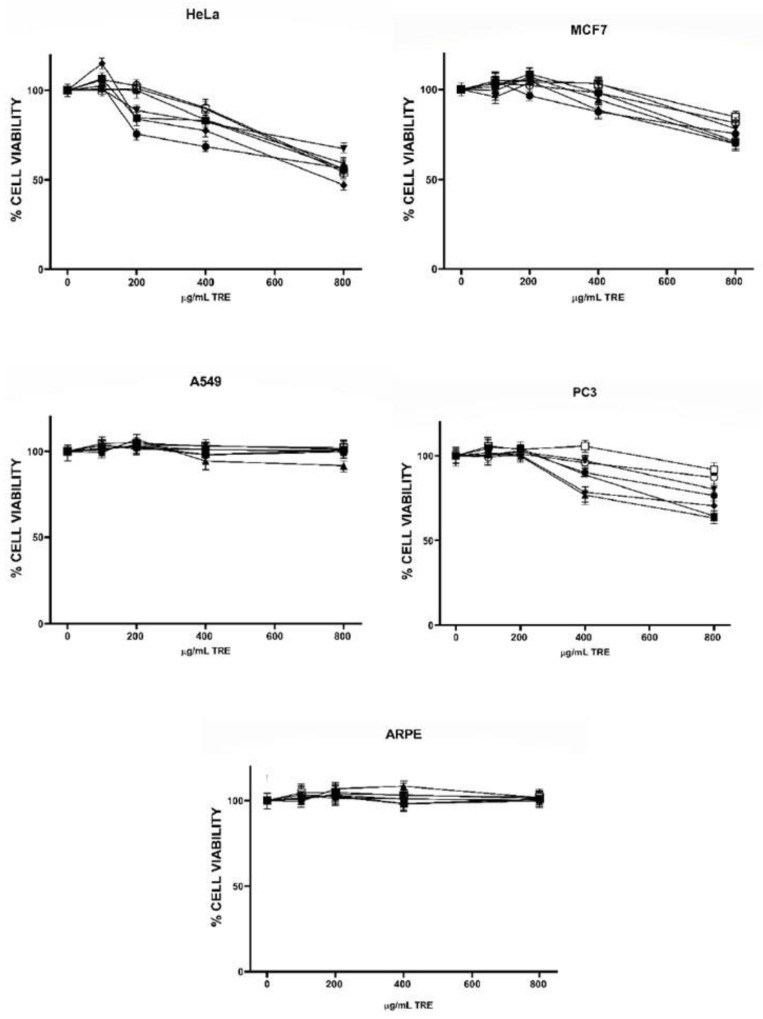
Effect of tocol-rich extracts (TRE) on viability cancer cells. Almond (ALM
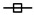
), Emory oak acorn (EOA
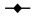
), Pecans (PEC
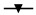
), Pine nut pink (PPN
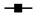
), Pine nut white (PNW
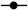
), Pistachio (PIS
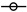
), Walnut (WN
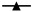
).

**Figure 2 molecules-27-03154-f002:**
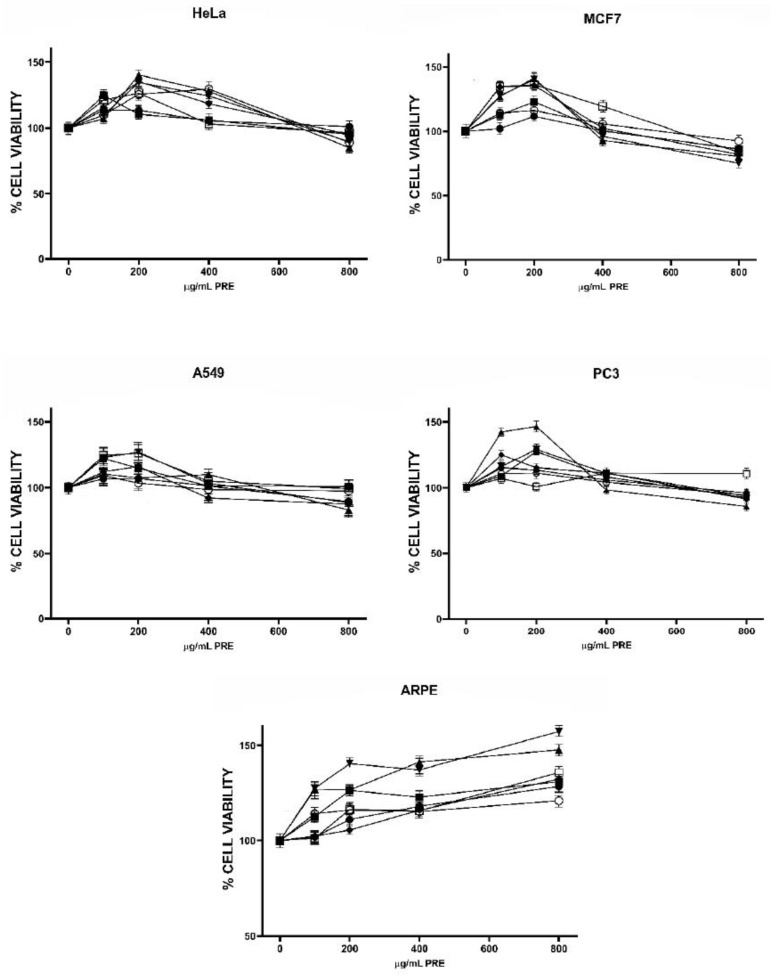
Effect of phenols-rich extracts (PRE) on viability cancer cells. Almond (ALM
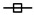
), Emory oak acorn (EOA
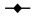
), Pecans (PEC
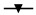
), Pine nut pink (PPN
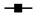
), Pine nut white (PNW
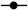
), Pistachio (PIS
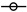
), Walnut (WN
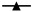
).

**Figure 3 molecules-27-03154-f003:**
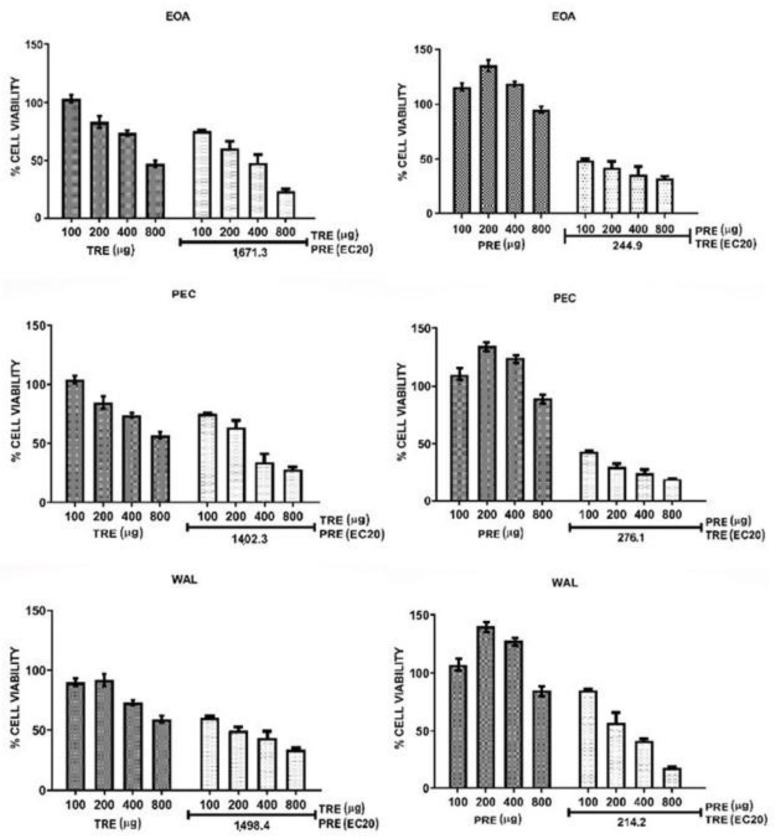
HeLa cell viability alone and in combination of EC20PRE: TRE (**left**) and EC20TRE: PRE (**Right**).

**Figure 4 molecules-27-03154-f004:**
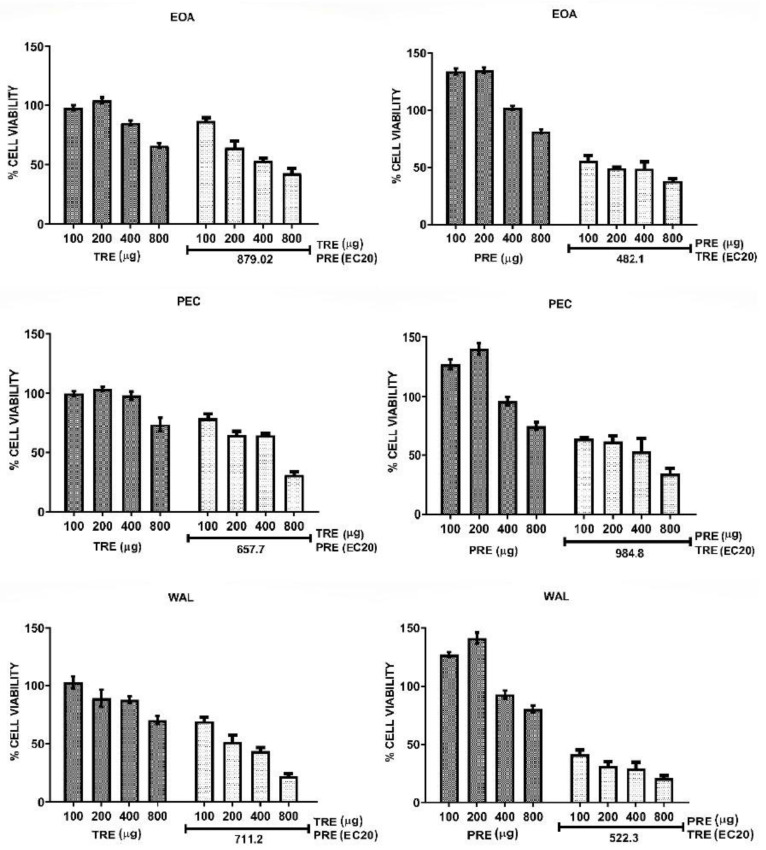
MCF7 cell viability alone and in combination of EC20PRE: TRE (**left**) and EC20TRE: PRE (**Right**).

**Figure 5 molecules-27-03154-f005:**
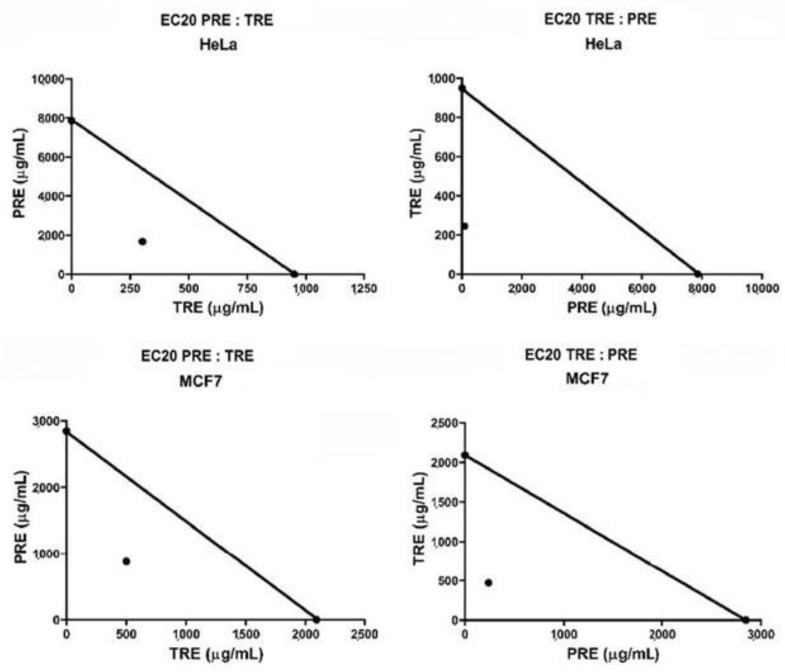
Isobologram of EC20PRE: TRE (**left**) and EC20TRE: PRE (**Right**) from EOA in HeLa (**up**) and MCF7 cells (**down**).

**Table 1 molecules-27-03154-t001:** Effect of pure compounds on the viability of cancer cells.

	HeLa	MCF7	A549	PC3
	EC50 µM	EC50 µM	EC50 µM	EC50 µM
αT	764.9 ± 19.2 ^c^ _i_	631.5 ± 12.3 ^e^ _i_	969.8 ± 78.8 ^c^ _i_	866.3 ± 106.2 ^c^ _i_
γT	659.2 ± 16.9 ^c^ _ii_	438.3 ± 8.0 ^d^ _i_	463.6 ± 14.7 ^b^ _i_	437.1 ± 0.3 ^b^ _i_
δT	386.7 ± 6.0 ^b^ _ii_	319.9 ± 9.8 ^c^ _i,ii_	491.9 ± 21.0 ^b^ _iii_	436.9 ± 11.6 ^b^ _ii,iii_
Gallic acid	281.4 ± 44.0 ^b^ _i_	151.3 ± 23.4 ^b^ _i_	421.9 ± 42.7 ^ab^ _i_	98.6 ± 32.6 ^ab^ _i_
Doxorubicin	2.36 ± 0.08 ^a^ _i_	2.30 ± 0.03 ^a^ _i_	4.7 ± 0.05 ^a^ _ii_	7.00 ± 0.5 ^a^ _iii_

α (αT), γ (γT), δ (δT), gallic acid and doxorubicin (Mean ± SEM). Each EC50 was calculated using a linear equation by log (dose) response curves. Different roman numerals in the same row indicate different significant values between cancer cell lines for each compound (*p* < 0.05). Different letters in the same column indicate different significant values between compounds for each cancer cell line (*p* < 0.05).

**Table 2 molecules-27-03154-t002:** Content of individual tocoferols (T) and tocotrienols (T3) in TRE extracts from nuts.

Nuts	αT	γT	δT	αT3	γT3	TT
Almond	43.4 ± 0.5 ^a^	318.3 ± 11.4 ^c^	97.38 ± 0.0 ^b^	ND	ND	459.1 ± 4.5 ^b^
Emory oak acorn	1.52 ± 0.1 ^c^	7170.5 ± 1227.1 ^a^	1695 ± 0.3 ^a^	UQL	219.7 ± 26.7 ^c^	9086.8 ± 250.0 ^a^
Pecan	0.05 ± 0.0 ^c^	564.0 ± 20.5 ^bc^	204.2 ± 0.2 ^b^	0.3 ± 0.0 ^b^	UQL	768.3 ± 93.0 ^b^
Pine nut, pink	7.79 ± 2.3 ^b^	3235.5 ± 807.1 ^abc^	818.2 ± 0.2 ^ab^	1740.5 ± 445.1 ^a^	793.4 ± 193.0 ^b^	6595 ± 164.0 ^ab^
Pine nut, white	0.18 ± 0.0 ^c^	4538.0 ± 38.9 ^abc^	UQL	136.3 ± 4.7 ^b^	1303.9 ± 1.2 ^a^	5978 ± 12.3 ^ab^
Pistachio	0.02 ± 0.0 ^c^	429.5 ± 2.4 ^c^	97.1 ± 0.0 ^b^	44.2 ± 3.4 ^b^	93.5 ± 0.0 ^c^	664.3 ± 4.6 ^b^
Walnut	ND	5225.1 ± 542.1 ^ab^	827.1 ± 0.3 ^ab^	ND	ND	6052.4 ± 207.1 ^ab^

UQL: Under quantification limit, ND: not detected. Values are presented as mean ± SEM (*n* = 3). Different letters in the same column indicates values significantly different between nuts (*p* < 0.05). Tocol rich extract (TRE), Tocopherols (T), tocotrienols (T3) and total tocols (TT) were expressed as micrograms of tocols per gram of extract (µg tocols/g of TRE).

**Table 3 molecules-27-03154-t003:** Content of total phenolic compounds in PRE from nuts.

Nuts	µg GAE/g PCE
Almond	512.5 ± 31.0 ^b^
Emory oak acorn	475.1 ± 16.3 ^b^
Pecan	823.4 ± 31.1 ^a^
Pine nut, pink	369.7 ± 0.8 ^b^
Pine nut, white	383.6 ± 11.3 ^b^
Pistachio	141.6 ± 2.91 ^c^
Walnut	717.9 ± 72.2 ^ab^

Values are presented as mean ± SEM (*n* = 3). Different letters in the same column indicates values significantly different (*p* < 0.05). Phenolic compounds rich extract (PCE) was expressed as micrograms of gallic acid equivalents (GAE) per grams of extract (µg GAE).

**Table 4 molecules-27-03154-t004:** Pearson correlation coefficients between content of bioactive compounds and cell viability in TRE- and PRE-treated cell cultures.

Compound in Extract	Correlation Coefficient ® with % Cell Viability of Cell Lines				
	HeLa	MCF7	A549	PC3	*ARPE*
αT	0.1995	0.5733	0.4529	0.5563	0.5060
γT	−0.7481	**−0.9071 ***	−0.6761	**−0.7696 ***	−0.5662
δT	**−0.8644 ***	**−0.8515 ***	−0.2293	−0.6346	−0.6466
Total T	**−0.7975 ***	**−0.9277 ***	−0.6135	**−0.7699 ***	−0.6004
αT3	0.2925	−0.2807	−0.2215	−0.4657	−0.0666
γT3	0.2229	−0.2377	**−0.8048 ***	−0.3382	−0.2324
Total T3	0.3063	−0.3061	−0.5591	−0.4790	−0.1631
Total T + T3	−0.6426	**−0.9336 ***	−0.7176	**−0.8380 ***	−0.5944
Total phenols	−0.5172	**−0.9715 ***	−0.6034	−0.1582	0.9778

Individual T and T3 were quantified in the TRE; total phenols were quantified in the PRE. **Bold *** indicates significant correlation (*p* < 0.05).

**Table 5 molecules-27-03154-t005:** Combination index (CI) to evaluate the interaction between TRE and PRE in cancer cell lines.

Nuts	CI for Combination TRE + PRE_EC20_		CI for Combination PRE + TRE_EC20_	
	HeLa	MCF7	HeLa	MCF7
EOA	0.53 (synergy)	0.55 (synergy)	0.12 (strong synergy)	0.30 (synergy)
PEC	0.44 (synergy)	0.37 (synergy)	0.18 (strong synergy)	0.28 (strong synergy)
WAL	0.52 (synergy)	0.43 (synergy)	0.27 (strong synergy)	0.22 (strong synergy)

## Data Availability

The data presented in this study is openly available in Mendeley Data, Stevens Barron, Jazmin (2022), “Synergistic interactions between tocol and phenolic extracts from different tree nut species against human cancer cell lines”, doi: 10.17632/4n4rjp9pdk.1.

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
