# Peer review of "Synergistic Interactions between Tocol and Phenolic Extracts from Different Tree Nut Species against Human Cancer Cell Lines"

_molecules, 2022, doi:10.3390/molecules27103154_

Round 1

Reviewer 1 Report

The content of the MS is interesting and well presented. Thanks.

Author Response

Dear Reviewer, thank you for reading our manuscript and for sending us your comments. We hope that the final version of the manuscript will be suitable for publication in Molecules.

Reviewer 2 Report

The data presented in the current manuscript are not consistent with published ones. 

Author Response

Dear Reviewer, thank you for reading our manuscript and for sending us your comments and suggestions. Our responses are given in a point-by-point manner below. We have highlighted the changes to our manuscript within the document by using Track Changes in MS Word.

  1. Line 22 question marks and highlighted in sentence: as they are present in the nut

Response: Thanks for your observation. The sentence was rewritten for better understanding: however, the combined effect of both types of compounds has been scarcely studied, and this approach could give valuable information on the real anticancer potential of tree nuts

  1. Line 52, please rewrite this.

Response: Thank you with your observation We have rewritten the sentence: Tree nuts are an example of this type of food [3–5]. Moreover, tree nuts are remarkable because, since they are composed of an oily fraction and a polar fraction, they possess both lipophilic and hydrophilic bioactive compounds.

  1. Line 60 thereby blocking the phases of cell growth and proliferation.

Response: We agree with your observation. The word thereby was added to the sentence They can modulate and inhibit estrogen receptors [7] and cyclin D1 [8] thereby blocking the phases of cell growth and proliferation

  1. Line 81, Chou-Talalay method

Response: Thanks for the observation, the name Chou-Talay was corrected by Chou-Talalay.

  1. Line 84, Chou-Talalay method

Response: Thanks for the observation, the name Chou-Talalay was corrected.

  1. Line 89, question marks, paragraph was rewritten for better understanding.

Response: We appreciate your comment; the paragraph was rewritten, and the citation corrected.

  1. Line 101, highlighted text in word y.

Response: Thanks for your observation, the word y is replaced by the symbol gamma.

  1. Table 1, highlighted text in word doxorrubicin.

Response: Thanks for your observation, the word doxorubicin was corrected in the table.

  1. Line 120, highlighted text in sentence (50 and 40 μM)

Response: Thank you for your observation, we have corrected the sentence for a better understanding of the paragraph.

No studies were found that determined the EC50 of tocopherols in HeLa cells. While the study by Jiang et al. [28] in PC3 and A549 cells also observed a dose response effect of γT, with EC50 values of 50 and 40 µM, respectively.

Once again, we thank you for taking the time to carefully read our manuscript and for making all these observations. We hope that our responses are satisfactory and that you find the revised version of the manuscript suitable for publication.

Reviewer 3 Report

The study “Synergistic interactions between tocol and phenolic extracts from different tree nut species against human cancer cell lines” has investigated anti-cancer effects of tocol- and phenolic-rich extracts from different tree nuts and focused on the synergistic interactions. The results are well presented in a rational way and contain some interesting information that could be useful in making advances in scientific fields. However, some editing is required which are listed below.

  1. Lines 32-33: “Combination of both extracts showed a synergistic and strongly synergistic behavior in the three nuts”. Please specify here the specific type of the nuts.
  2. Lines 55-56: ”Ts, mainly αT, are the main forms of vitamin E, they are effective antioxi- 55dants, and the γ and δ forms also possess anticancer effects.” Please add reference to this sentence.
  3. Lines 78-82: Type of different tree nut species and some detailed information regarding the selected species is suggested to be supplemented in aim of this study.
  4. Lines 126-127: “γT was the most abundant isoform in all TRE and was highest in EOA; EOA, PNP and WAL extracts”. This is the first time EOA, PNP and WAL was mentioned in the main text, please clarify those abbreviations.
  5. Fig.1 and Fig.2: Standard error bars were missing in the two figures. Please supplement.
  6. Lines 202-204: Result of effect of PRE on viability cancer cells (Fig.2) is surprisingly high, and the author attributed this result to discrepancies between PRE concentrations. Why not applying a higher concentration in the current study, based on the odd observation?
  7. Lines 223-224: “EC50 values of all TRE in combination with PRE were lower than 800 µg/mL and higher than 800 µg/mL if used alone.” This sentence is difficult to be understood, please revise.
  8. Lines 224-226: Please add some detailed data to describe this result.
  9. Figure 5. Please reduce the font of Ec20PRE:TRE and Ec20TRE:PRE on the top of the figure.

Author Response

Dear Reviewer, thank you for reading our manuscript and for sending us your comments and suggestions. Our responses are given in a point-by-point manner below. We have highlighted the changes to our manuscript within the document by using Track Changes in MS Word.

  1. Lines 32-33: “Combination of both extracts showed a synergistic and strongly synergistic behavior in the three nuts”. Please specify here the specific type of the nuts.

Response: Thank you for your observation, we have specified the nuts with the synergistic effect.

  1. Lines 56-57: ”Ts, mainly αT, are the main forms of vitamin E, they are effective antioxidants, and the γ and δ forms also possess anticancer effects.” Please add reference to this sentence.

Response: Thank you for your comment, we add these two references to the sentence:

Smolarek, A.K.; So, J.Y.; Burgess, B.; Kong, A.T.; Reuhl, K.; Lin, Y.; Shih, W.J.; Li, G.; Lee, M.; Chen, Y. Dietary administration of δ- and γ -tocopherol inhibits tumorigenesis in the animal model of estrogen receptor – positive , but not HER-2 breast cancer. 2012, 5, 1310–1321, doi:10.1158/1940-6207.CAPR-12-0263.

Bak, M.J.; Gupta, S. das; Wahler, J.; Lee, H.J.; Li, X.; Lee, M.J.; Yang, C.S.; Suh, N. Inhibitory effects of γ- and δ-tocopherols on estrogen-stimulated breast cancer in vitro and in vivo. Cancer Prevention Research 2017, 10, 188–197, doi:10.1158/1940-6207.CAPR-16-0223.

  1. Lines 83-84: Type of different tree nut species and some detailed information regarding the selected species is suggested to be supplemented in aim of this study.

Response: We agree with your observation, the detailed information on the selected nuts was added in the aim of the study, the abbreviations were also added in this paragraph:

For this purpose, seven tree nut species were selected. Six of them are commercial and commonly consumed as healthy snacks: almond (ALM), pecan (PEC), pine nuts from two varieties (pink [PNP] and white [PNW]), pistachio (PIS) and walnut (WAL). The last one is a wild emory oak acorn (EOA) species characteristic of Northern Mexico, which can be used as an ingredient in baking products. Since each tree nut species possess a unique profile of tocols and polyphenols, their antiproliferative potential will be compared among them.

  1. Lines 144-145: “γT was the most abundant isoform in all TRE and was highest in EOA; EOA, PNP and WAL extracts”. This is the first time EOA, PNP and WAL was mentioned in the main text, please clarify those abbreviations.

Response: We appreciate your observation; the abbreviations were added in the introduction.

  1. 1 and Fig.2: Standard error bars were missing in the two figures. Please supplement.

Response: Standard error bars were added in the Figure 1 and Figure 2.

  1. Lines 226-227: Result of effect of PRE on viability cancer cells (Fig.2) is surprisingly high, and the author attributed this result to discrepancies between PRE concentrations. Why not applying a higher concentration in the current study, based on the odd observation?

Response: Yes, as indicated in the manuscript some studies have found antiproliferative activity of phenolic extracts at mg/mL, however in our lab we usually study the antiproliferative effects using 800 µg/mL as a maximum concentration in order to find more potent bioactive compounds. For that reason, instead of increasing the extract concentration we sought to improve its efficacy by combining it with the TRE.

  1. Lines 232-233: “EC50 values of all TRE in combination with PRE were lower than 800 µg/mL and higher than 800 µg/mL if used alone.” This sentence is difficult to be understood, please revise.

Response: Thank you for your observation, the sentence was rewritten for a better understanding The combination of TRE + PREEC20 had a greater antiproliferative effect than TRE alone, all EC50 values of all TRE in combination with PRE were lower than 800 μg/mL, unlike when tested alone on cells, this means that combined extracts are more effective than extracts alone.

  1. Lines 234-235: Please add some detailed data to describe this result.

Response: We have added some detailed data and restructured the paragraph for better understanding. For the combinations PRE+ TREEC20 the effect was more remarkable in HeLa and MCF7, the EC50 of the combination always remained below 500 µg (Table 3 in Supplementary Data), and the combinations were more effective than PRE alone [EC50 above 2.0 milligrams (Table 3 in Supplementary Data) ; moreover, combination treatments did not induce increase in viability at low PRE concentrations (200 µg), as observed when PRE were used alone (see Figure 3 and Figure 4).

  1. Figure 5. Please reduce the font of Ec20PRE:TRE and Ec20TRE:PRE on the top of the figure.

Response: Thanks for your comment, the source of Ec20PRE:TRE and Ec20TRE:PRE has been reduced at the top of the figure.

Once again, we thank you for taking the time to carefully read our manuscript and for making all these observations. We hope that our responses are satisfactory and that you find the revised version of the manuscript suitable for publication.